# PRVAE-VC: Non-Parallel Many-to-Many Voice Conversion with Perturbation-Resistant Variational Autoencoder

*Kou Tanaka, Hirokazu Kameoka, Takuhiro Kaneko*

NTT Communication Science Laboratories, NTT Corporation, Japan

{kou.tanaka.ef, hirokazu.kameoka.uh, takuhiro.kaneko.tb}@hco.ntt.co.jp

## Abstract

This paper describes a novel approach to non-parallel many-to-many voice conversion (VC) that utilizes a variant of the conditional variational autoencoder (VAE) called a perturbation-resistant VAE (PRVAE). In VAE-based VC, it is commonly assumed that the encoder extracts content from the input speech while removing source speaker information. Following this extraction, the decoder generates output from the extracted content and target speaker information. However, in practice, the encoded features may still retain source speaker information, which can lead to a degradation of speech quality during speaker conversion tasks. To address this issue, we propose a perturbation-resistant encoder trained to match the encoded features of the input speech with those of a pseudo-speech generated through a content-preserving transformation of the input speech's fundamental frequency and spectral envelope using a combination of pure signal processing techniques. Our experimental results demonstrate that this straightforward constraint significantly enhances the performance in non-parallel many-to-many speaker conversion tasks. Audio samples can be accessed at our webpage [1].

**Index Terms**: Voice conversion, variational autoencoder, perturbation resistance, representation learning, non-parallel

## 1. Introduction

Voice conversion (VC) is a technique that transforms the speech of one speaker to sound like that of another while preserving linguistic content. This technique finds applications in various domains, including speaker conversion [1, 2], assistive systems [3, 4] aimed at overcoming speech and hearing impairments, and pronunciation and accent conversions [5] for language learning.

There are two frameworks for learning conversion models: parallel VC and non-parallel VC. Parallel VC [2, 6] requires a parallel speech corpus consisting of recordings of the same text spoken by both the source and target speakers. While collecting such a corpus can be time-consuming and expensive, it has the potential to produce high-quality results since it allows for direct optimization based on the target speech. In contrast, non-parallel VC involves converting the source speech to the target speech without explicitly aligning the source and target utterances. This makes the task more challenging, as the model has to learn the correspondence between the source and target speech without any guidance from parallel data. However, non-parallel VC has become an active research area in recent years due to the availability of a large amount of non-parallel speech data.

There are two primary methodologies for developing non-parallel VC: one involving text supervision and the other being unsupervised. Non-parallel VC using text supervision [7, 8] is also known as an approach cascading automatic speech recognition (ASR) and text-to-speech synthesis (TTS). It utilizes a phoneme recognizer to extract phonetic information from the input speech, which is then fed to TTS to generate the output speech. While this approach can produce high-quality conversion results if the ASR works well, it requires paired data of text and speech for training ASR and TTS, which can be a limiting factor. In contrast, non-parallel VC without text supervision [9–11] typically employs techniques such as autoencoders (AE) [12], variational autoencoders (VAE) [13], and generative adversarial networks (GAN) [14]. This work focuses on non-parallel VC based on a VAE-based system without text supervision, as it has the potential to utilize latent space to represent common hidden features of speech signals among different speakers.

The VAE-based VC [15] employs a latent space typically assumed to follow a Gaussian distribution to encode a set of input acoustic features such as Mel-spectrogram. Then, the speaker information is added to the encoded latent features in the generation phase to obtain the output acoustic features. In the decoder, the source speaker information is used to estimate the reconstruction of the input acoustic features, while that of the target speaker is used to estimate the converted acoustic features. Although speaker conversion can be achieved by setting the appropriate hyperparameters, such as the number of dimensions of the model, various improvements have been proposed to achieve better conversion. An example of such an approach is cycle consistency [16], which ensures that the converted speech can be converted back to its original form with the output being as close to the original speech as possible. Another variation involves incorporating an auxiliary classifier [10], which prevents the decoder from disregarding the speaker information. Unfortunately, according to our initial experiments, these variants still suffer from hyperparameter tuning of the model. There is a large difference in conversion performance between small and large model sizes. One possible reason is that the latent space is not uniform across all speakers, and as the model size expands, it forms different distributions for each speaker to match the training data better.

To address this issue, we propose a variant of the conditional VAE called a perturbation-resistant VAE (PRVAE). In our approach, a perturbation-resistant encoder is trained to match the encoded features of the input speech with those of a pseudo-speech. The pseudo-speech is generated by applying content-preserving transformations to the input speech using pure signal processing techniques. This work defines content-preserving transformations as linear transformations of fundamental fre-

---

[1] http://www.kecl.ntt.co.jp/people/tanaka.ko/projects/prvaevc/

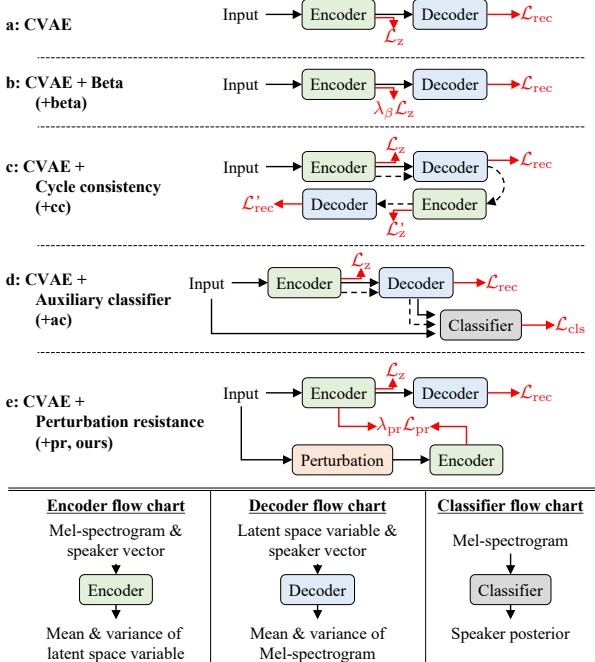

**a: CVAE**
Input → Encoder → Decoder → $\mathcal{L}_{\text{rec}}$
$\mathcal{L}_{\text{z}}$

**b: CVAE + Beta (+beta)**
Input → Encoder → Decoder → $\mathcal{L}_{\text{rec}}$
$\lambda_\beta \mathcal{L}_{\text{z}}$

**c: CVAE + Cycle consistency (+cc)**
Input → Encoder → Decoder → $\mathcal{L}_{\text{rec}}$
$\mathcal{L}_{\text{z}}$
Decoder ← Encoder
$\mathcal{L}'_{\text{rec}}$ $\mathcal{L}'_{\text{z}}$

**d: CVAE + Auxiliary classifier (+ac)**
Input → Encoder → Decoder → $\mathcal{L}_{\text{rec}}$
$\mathcal{L}_{\text{z}}$
Classifier → $\mathcal{L}_{\text{cls}}$

**e: CVAE + Perturbation resistance (+pr, ours)**
Input → Encoder → Decoder → $\mathcal{L}_{\text{rec}}$
$\mathcal{L}_{\text{z}}$
$\lambda_{\text{pr}} \mathcal{L}_{\text{pr}}$
Perturbation → Encoder

| **Encoder flow chart** | **Decoder flow chart** | **Classifier flow chart** |
|---|---|---|
| Mel-spectrogram & speaker vector | Latent space variable & speaker vector | Mel-spectrogram |
| ↓ | ↓ | ↓ |
| Encoder | Decoder | Classifier |
| ↓ | ↓ | ↓ |
| Mean & variance of latent space variable | Mean & variance of Mel-spectrogram | Speaker posterior |

Figure 1: *System overview of VAE-based VCs. Black solid and dashed arrows in (a)-(e) indicate the reconstruction and conversion flow. The solid red arrow indicates the loss calculation.*

quency and spectral envelope without changing the linguistic content. Our experimental results demonstrate that introducing perturbation resistance successfully overcomes the unstable behavior caused by changes in model parameters. This finding proves that increasing the model size can improve performance, as shown in subjective and objective evaluations.

## 2. Conventional VAE-Based VC

The system overview is shown in Fig. 1. We only require the speech waveform and the corresponding speaker ID as the training data.

As the speech parameters, we extract 80-dimensional Mel-spectrogram features over a range of 80-7600 Hz from the given source speech signals sampled at 16 kHz. The requirements for short-time Fourier transform are the same as reported in [17]; a Hanning window, 64 ms frame length, eight ms frameshift, and 1024-point fast Fourier transform. Instead of using classical vocoders such as STRAIGHT [18] or WORLD [19], which were used in some conventional methods, we used HiFi-GAN [20], a neural vocoder, to synthesize speech waveforms. To ensure a fair comparison of all methods, in our experiment, we 1) extracted the Mel-spectrogram from the speech waveform, 2) converted the Mel-spectrogram using each method, and 3) finally generated the speech waveform using HiFiGAN.

### 2.1. Conditional VAE (CVAE)

A conditional variant [21] of VAE [13] is a neural network model that includes an encoder network and a decoder network. The encoder network produces parameters for the conditional distribution $q_\phi(z|x, c)$ of a latent space variable $z$, given data $x$ and the attribute codes $c$. In contrast, the decoder network generates parameters for the conditional distribution $p_\theta(x|z, c)$

of the data $x$, given the latent space variable $z$ and the attribute codes $c$. The log marginal distribution of the data $x$, given the attribute codes $c$, is given as:

$$\log p_\theta(\boldsymbol{x}|\boldsymbol{c}) = \mathcal{L}(\theta, \phi) + \text{KL}[q_\phi(\boldsymbol{z}|\boldsymbol{x}, \boldsymbol{c})|p(\boldsymbol{z})], \quad (1)$$

where $\text{KL}[\cdot|\cdot]$ denotes the Kullback-Leibler (KL) divergence. This implies we can minimize the KL divergence between $q_\phi(z|x, c)$ and $p(z)$ by maximizing $\mathcal{L}(\theta, \phi)$ with respect to $\theta$ and $\phi$. A typical way of modeling $p(z)$, $q_\phi(z|x, c)$, and $p_\theta(x|z, c)$, is to assume Gaussian distributions.

In the conditional VAE (CVAE) based VC [15], the encoder and decoder networks are designed to generate the sequences of the means and logarithmic variances of $q_\phi$ and $p_\theta$, given the Mel-spectrogram $x_s$ and the speaker codes $c_s$ of the source speaker:

$$\left[\boldsymbol{\mu}_{\boldsymbol{z}_s}; \log \boldsymbol{\sigma}_{\boldsymbol{z}_s}^2\right] = \text{Encoder}(\boldsymbol{x}_s, \boldsymbol{c}_s), \quad (2)$$

$$\left[\boldsymbol{\mu}_{\boldsymbol{x}_{ss}}; \log \boldsymbol{\sigma}_{\boldsymbol{x}_{ss}}^2\right] = \text{Decoder}(\boldsymbol{\mu}_{\boldsymbol{z}_s} + \boldsymbol{\sigma}_{\boldsymbol{z}_s} \odot \boldsymbol{\epsilon}, \boldsymbol{c}_s), \quad (3)$$

$$\boldsymbol{x}_{ss} = \boldsymbol{\mu}_{\boldsymbol{x}_{ss}} + \boldsymbol{\sigma}_{\boldsymbol{x}_{ss}} \odot \boldsymbol{\epsilon}, \quad (4)$$

where $\epsilon$, $[;]$, and $\odot$ denotes Gaussian noise, concatenation along the channel dimension, and element-wise manipulation. In the conversion process at the test time, given the speaker codes $c_t$ of the target speaker, the converted Mel-spectrogram $x_{st}$ is generated as follows:

$$\left[\boldsymbol{\mu}_{\boldsymbol{x}_{st}}; \log \boldsymbol{\sigma}_{\boldsymbol{x}_{st}}^2\right] = \text{Decoder}(\boldsymbol{\mu}_{\boldsymbol{z}_s} + \boldsymbol{\sigma}_{\boldsymbol{z}_s} \odot \boldsymbol{\epsilon}, \boldsymbol{c}_t), \quad (5)$$

$$\boldsymbol{x}_{st} = \boldsymbol{\mu}_{\boldsymbol{x}_{st}} + \boldsymbol{\sigma}_{\boldsymbol{x}_{st}} \odot \boldsymbol{\epsilon}. \quad (6)$$

Finally, the objective function $\mathcal{L}_{\text{cvae}}$ to be minimized is given as,

$$\mathcal{L}_{\text{cvae}} = \mathcal{L}_{\text{z}} + \mathcal{L}_{\text{rec}}, \quad (7)$$

$$\mathcal{L}_{\text{z}} = F_{\text{KLD}}\left(\mathcal{N}(\boldsymbol{\mu}_{\boldsymbol{x}_{ss}}, \boldsymbol{\sigma}_{\boldsymbol{x}_{ss}}^2)|\mathcal{N}(\boldsymbol{0}, \boldsymbol{I})\right),, \quad (8)$$

$$\mathcal{L}_{\text{rec}} = F_{\text{GNLL}}\left(\boldsymbol{x}_s, \mathcal{N}(\boldsymbol{\mu}_{\boldsymbol{x}_{ss}}, \log \boldsymbol{\sigma}_{\boldsymbol{x}_{ss}}^2)\right), \quad (9)$$

where $\mathcal{N}(\cdot)$, $F_{\text{GNLL}}$, and $F_{\text{KLD}}$ denote a Gaussian distribution, a Gaussian negative log-likelihood loss function, and a KL divergence loss function, respectively. As shown in Fig. 1(a), the well-known KL loss for the latent space variable and reconstruction loss for the data are Eqs. (8) and (9).

### 2.2. Beta variant

Beta-VAE [22] is a variant of the VAE model that emphasizes the disentanglement of the latent space variables. In a typical VAE, the latent space variables $z$ follow a multidimensional Gaussian distribution. However, the constraint may be weakened due to the balance between the KL divergence term $\mathcal{L}_{\text{z}}$ and the reconstruction error $\mathcal{L}_{\text{rec}}$ in the objective function, Eq (7). Beta-VAE strengthens the constraint by increasing the weight $\lambda_{\text{beta}}$ of the KL term to more than 1, promoting independence and disentanglement of the latent space variables across dimensions (Fig. 1(b)). However, this weakens the importance of the reconstruction error, potentially resulting in blurred reconstructed data. The objective functions of Beta-VAE to be minimized is given as,

$$\mathcal{L}_{+\text{beta}} = \lambda_{\text{beta}} \mathcal{L}_{\text{z}} + \mathcal{L}_{\text{rec}}. \quad (10)$$

Note that [23] uses Beta-VAE to model both speaker and content information in the encoder. In contrast, we explicitly incorporate speaker information using speaker IDs to compare with other methods under the same conditions.

## 2.3. Cycle-consistent variant

CycleVAE [16] is a variant of the VAE model that takes into account not only the reconstruction flow but also the conversion flow in the parameter optimization (Fig. 1(c)). As shown in Eq. (7), the original CVAE objective function consisted of flows to reconstruct the input $\boldsymbol{x}_s$ and did not consider the actual conversion process. To address this problem, [16] indirectly optimizes the conversion flow by recycling the converted features $\boldsymbol{x}_{st}$ back into the system to obtain corresponding cyclic reconstructed features $\boldsymbol{x}_{sts}$ that can be directly optimized, as follows:

$$\left[\boldsymbol{\mu}_{\boldsymbol{z}_{st}}; \log \boldsymbol{\sigma}_{\boldsymbol{z}_{st}}^2\right] = \text{Encoder}(\boldsymbol{x}_{st}, \boldsymbol{c}_t), \tag{11}$$

$$\left[\boldsymbol{\mu}_{\boldsymbol{x}_{sts}}; \log \boldsymbol{\sigma}_{\boldsymbol{x}_{sts}}^2\right] = \text{Decoder}(\boldsymbol{\mu}_{\boldsymbol{z}_{st}} + \boldsymbol{\sigma}_{\boldsymbol{z}_{st}} \odot \boldsymbol{\epsilon}, \boldsymbol{c}_s). \tag{12}$$

Since VAE is trained using unaligned speech data, the ground truth Mel-spectrogram for $\boldsymbol{x}_{st}$ does not exist in the training data. However, since $\boldsymbol{x}_{sts}$ is expected to be the input Mel-spectrogram $\boldsymbol{x}_s$, the losses $\mathcal{L}'_z$ and $\mathcal{L}'_{\text{rec}}$ can still be calculated, as follows:

$$\mathcal{L}'_z = F_{\text{KLD}}\left(\mathcal{N}(\boldsymbol{\mu}_{\boldsymbol{x}_{st}}, \boldsymbol{\sigma}_{\boldsymbol{x}_{st}}^2) | \mathcal{N}(\boldsymbol{0}, \boldsymbol{I})\right), \tag{13}$$

$$\mathcal{L}'_{\text{rec}} = F_{\text{GNLL}}\left(\boldsymbol{x}_s, \mathcal{N}(\boldsymbol{\mu}_{\boldsymbol{x}_{sts}}, \log \boldsymbol{\sigma}_{\boldsymbol{x}_{sts}}^2)\right). \tag{14}$$

This cyclic flow can be continued by using the cyclic reconstructed features $\boldsymbol{x}_{sts}$ as input $\boldsymbol{x}_s$ for the next cycle. The objective functions $\mathcal{L}_{+cc}$ of CycleVAE to be minimized is given as,

$$\mathcal{L}_{+cc} = \frac{1}{N_{cc}} \sum_{N_{cc}} (\mathcal{L}_z + \mathcal{L}_{\text{rec}} + \mathcal{L}'_z + \mathcal{L}'_{\text{rec}}). \tag{15}$$

where $N_{cc}$ indicates the total number of cycle.

## 2.4. Auxiliary classifier variant

ACVAE [16] is a variant of the VAE model that considers both the reconstruction flow and the conversion flow in the parameter optimization process (Fig. 1(d)). Unlike [16], ACVAE employs information-theoretic regularization during model training to ensure that the information contained in the attribute class label is preserved in the conversion process. In a standard CycleVAE, the encoder and decoder networks can still disregard the attribute class labels. This results in limited control over the speech characteristics during testing, potentially leading to simple reconstruction without conversion.

To address this issue, ACVAE introduces an auxiliary classifier that takes the Mel-spectrogram $\boldsymbol{x}_s$, $\boldsymbol{x}_{ss}$, and $\boldsymbol{x}_{st}$ as input and estimates the logits $\boldsymbol{y}_s$, $\boldsymbol{y}_{ss}$, and $\boldsymbol{y}_{st}$ of the speaker posteriors as output, as follows:

$$\boldsymbol{y}_s = \text{Classifier}(\boldsymbol{x}_s), \tag{16}$$

$$\boldsymbol{y}_{ss}, \; \boldsymbol{y}_{st} = \text{Classifier}(\boldsymbol{x}_{ss}), \; \text{Classifier}(\boldsymbol{x}_{st}). \tag{17}$$

This enables us to optimize the conversion flow directly by learning the encoder, decoder, and classifier. The objective functions $\mathcal{L}_{+ac}$ of ACVAE to be minimized is given as,

$$\mathcal{L}_{+ac} = \mathcal{L}_{\text{cvae}} + \mathcal{L}_{\text{cls}_{\text{real}}} + \mathcal{L}_{\text{cls}_{\text{fake}}}, \tag{18}$$

$$\mathcal{L}_{\text{cls}_{\text{real}}} = F_{\text{CE}}(\boldsymbol{y}_s, \boldsymbol{h}_s), \tag{19}$$

$$\mathcal{L}_{\text{cls}_{\text{fake}}} = 0.5 * \left(F_{\text{CE}}(\boldsymbol{y}_{ss}, \boldsymbol{h}_s) + F_{\text{CE}}(\boldsymbol{y}_{st}, \boldsymbol{h}_t)\right). \tag{20}$$

where $F_{\text{CE}}$, $\boldsymbol{h}_s$, and $\boldsymbol{h}_t$ denote a cross-entropy loss function, the index of the source speaker, and that of the target speaker, respectively.

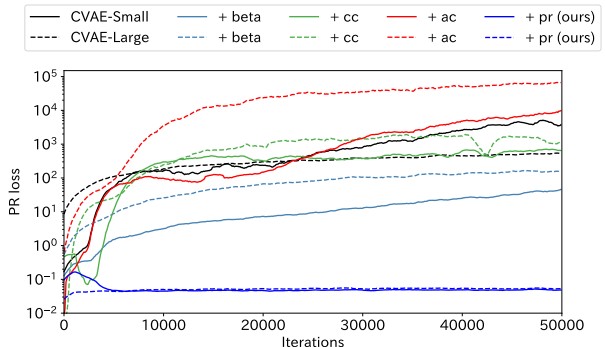

Figure 2: *Comparison of perturbation resistance losses on conventional VAE-based voice conversions and the proposed. Solid and dashed lines indicate the results on the small and large models, respectively.*

# 3. Proposed Method

## 3.1. Concept

It is a common assumption that the encoder in voice conversion extracts the content from the input speech while eliminating any information about the source speaker. For example, in cascading ASR and TTS approaches, ASR, which can be regarded as the encoder, extracts context information, a speaker-independent feature. The conventional VAE-based VCs introducing speaker codes are also assumed that the latent space variables, the output of the encoder, are the speaker-independent features expected to represent phonetic information [16]. After this extraction, the decoder uses the extracted content and target speaker information to generate the output Mel-spectrogram.

As a preliminary experiment to confirm the speaker independence of latent space variables, we calculated the KL divergence between the conditional distribution obtained when a certain Mel-spectrogram was given to the encoder and the conditional distribution obtained when a Mel-spectrogram of the pseudo-speech was given to the encoder, in which the mean of fundamental frequency ($F_0$) was randomly changed. Since the role of the encoder is to remove speaker bias, the KL divergence mentioned above should be close to zero, as the difference in the mean of $F_0$ can be considered a form of speaker bias. However, as shown in Fig. 2, the results of the KL divergence are quite large, indicating that the encoded features may still retain source speaker information. This could lead to a degradation of speech quality during speaker conversion tasks. To address this issue, we propose a training framework to learn less speaker-dependent features as the latent space variables without text supervision.

## 3.2. Perturbation-resistant VAE

To learn a speech representation that is less speaker-dependent in an unsupervised manner, pseudo-speech that manipulates speaker biases such as the mean value of $F_0$ and vocal tract length is created and used for training. Inspired by [24], we use WOLRD analyzer $F_{\text{ana}}$ and synthesizer $F_{\text{syn}}$ [19] to extract $F_0$ $\boldsymbol{f}_s$, spectral envelopes $\boldsymbol{e}_s$, and aperiodicities $\boldsymbol{a}_s$ from the original speech $\boldsymbol{w}_s$, and generate waveforms of pseudo-speech $\boldsymbol{w}_m$

from manipulated acoustic features, as follows:

$$\boldsymbol{f}_s, \boldsymbol{e}_s, \boldsymbol{a}_s = F_{\mathrm{ana}}(\boldsymbol{w}_s), \tag{21}$$

$$\boldsymbol{w}_m = F_{\mathrm{syn}}\big(F_{\mathrm{f0}}(\boldsymbol{f}_s, \alpha_f), F_{\mathrm{env}}(\boldsymbol{e}_s, \alpha_e), \boldsymbol{a}_s\big), \tag{22}$$

where $F_{\mathrm{f0}}$, $\alpha_f$, $F_{\mathrm{env}}$, and $\alpha_e$ represent a function that randomizes the mean of $F_0$, a target mean value for $F_0$, a frequency warping function [25], and a warping factor, respectively.

After generating the pseudo-speech, we extract the Mel-spectrogram $\boldsymbol{x}_m$ from $\boldsymbol{w}_m$ similarly to the extraction of $\boldsymbol{x}_s$ from $\boldsymbol{w}_s$. Unlike the conventional VAE-based VCs, a speaker encoder is introduced to obtain the speaker codes $\boldsymbol{c}_m$ from the Mel-spectrogram of the pseudo-speech. Then, a set of the parameters, $\boldsymbol{\mu}_{\boldsymbol{z}_m}$ and $\boldsymbol{\sigma}_{\boldsymbol{z}_m}$, for the conditional distribution of the latent space variable $\boldsymbol{z}_m$ is generated, as follow:

$$\boldsymbol{c}_m = \mathrm{SpeakerEncoder}(\boldsymbol{x}_m), \tag{23}$$

$$\big[\boldsymbol{\mu}_{\boldsymbol{z}_m}; \log \boldsymbol{\sigma}_{\boldsymbol{z}_m}^2\big] = \mathrm{Encoder}(\boldsymbol{x}_m, \boldsymbol{c}_m). \tag{24}$$

Our goal is to train the encoder to match the two distributions of the latent space variables $\boldsymbol{z}_s$ and $\boldsymbol{z}_m$. Hence, we define the perturbation resistance loss as follows:

$$\mathcal{L}_{\mathrm{pr}} = F_{\mathrm{KLD}}\big(\mathcal{N}(\boldsymbol{\mu}_{\boldsymbol{z}_s}, \boldsymbol{\sigma}_{\boldsymbol{x}_s}^2) | \mathcal{N}(\boldsymbol{\mu}_{\boldsymbol{z}_m}, \boldsymbol{\sigma}_{\boldsymbol{x}_m}^2)\big). \tag{25}$$

The final objective function $\mathcal{L}$ of PRVAE is given as,

$$\mathcal{L} = \mathcal{L}_{\mathrm{cvae}} + \lambda_{\mathrm{pr}} \mathcal{L}_{\mathrm{pr}}, \tag{26}$$

where $\lambda_{\mathrm{pr}}$ is a regularization parameter, which weighs the importance of the perturbation-resistant regularization.

# 4. Experiments

## 4.1. Implementation details

As detailed in Table 1, CVAEs were designed using long short-term memory (LSTM) [26]. To examine the conversion performance for different model sizes, we used two types of CVAEs in the experiment: Small and Large. Therefore, we evaluated 10 VC systems: **CVAE-Small**, 4 variants of CVAE-Small (**+beta**: beta, **+cc**: cycle-consistent, **+ac**: auxiliary classifier, and **+pr**: perturbation-resistant), **CVAE-Large**, and 4 variants of CVAE-Large (**+beta**, **+cc**, **+ac**, and **+pr**). In the CVAE-Small, 4-layer LSTMs with 128 hidden units were used for the encoder and decoder, respectively. In contrast, CVAE-Large uses a 2-layer LSTM with 512 hidden units, and the second layer of the stacked LSTMs has a residual connection. Speaker vectors were concatenated in all layers.

The model was trained for 100k iterations using the Adam optimizer [27] with a mini-batch size of 16. The learning rate, the first and second moments decay rates $\beta_1$, and $\beta_2$ were set to 0.001, 0.9, and 0.99, respectively. To train the small-sized models, we applied the KL term annealing technique [28], which gradually increases the weight of the KL divergence term in the objective function during training. This technique has been shown to improve the quality of generated samples and prevent the model from ignoring the latent variables. After experimenting with different values (2, 3, 5, and 10), we set $\lambda_\beta$ to 3. We also used $N_{cc}$ of 3, similar to the setting in [16]. The settings for the auxiliary classifier are the same as those in [16]. After experimenting with different values (1, 10, and 100), we set $\lambda_{\mathrm{pr}}$ to 10. As for the hyperparameters $\alpha_f$ and $\alpha_e$, we randomly sampled from uniform distributions of $[90, 300]$ and $[0.9, 1.1]$ for each training iteration, respectively. As the speaker encoder, we used 2-layer LSTMs with 128 hidden units, followed by a 32-dimensional linear projection to obtain a 32-dimensional speaker vector.

Table 1: *Model architecture summary for Small and Large VAE models.*

|  |  |  | Small | Large |
|---|---|---|---|---|
| Encoder | Projection | dim. | 128 | 512 |
|  | LSTM | layer | 4 | 2 |
|  |  | dim. | 128 | 512 |
|  |  | residual |  | ✓ |
|  | Projection | dim. | $16 \times 2$ | $32 \times 2$ |
| Decoder | Projection | dim. | 128 | 512 |
|  | LSTM | layer | 4 | 2 |
|  |  | dim. | 128 | 512 |
|  |  | residual |  | ✓ |
|  | Projection | dim. | $80 \times 2$ | |
| Num. of Params |  |  | 1.2M | 8.8M |

## 4.2. Other experimental conditions

We conducted experimental evaluations using a phonetically balanced Japanese speech dataset [29] consisting of utterances by six professional male speakers and four professional female speakers. The speech was recorded in a quiet room with minimal reverberation, and the silent section was removed using annotation labeled by experts. To train VC models, we used 450 sentences (speech section of around 0.5 hours) per speaker. To evaluate the performance, we used 53 sentences per speaker. All models were trained on *many-to-many* condition, which is 10-speaker input and 10-speaker output.

As the objective evaluation metrics, we used Mel-cepstral distortion (MCD) [dB] [30], a correlation coefficient of logarithmic $F_0$ ($F_0$Corr), and character error rate (CER) [%]. We used dynamic time warping [31] to get the alignment between the converted sample and the reference sample. To calculate the MCD and $F_0$Corr, we extracted 0-24 order Mel-cepstrum and $F_0$ from the raw speech and the converted speech synthesized by the neural vocoder. The CER was calculated by the Transformer-based ASR model trained on the corpus of spontaneous Japanese [32], provided by ESPnet [33]. Before calculating the CER, we converted kanji to hiragana to eliminate any variation caused by kanji or hiragana.

As the subjective evaluation of sound quality, we conducted a 5-scaled mean opinion score (MOS): 5 for excellent, 4 for good, 3 for fair, 2 for poor, and 1 for bad. To confirm speaker similarity, we also conducted a 4-scaled preference test (PT): 4 for same (sure), 3 for same (not sure), 2 for different (not sure), and 1 for different (sure). Ten native Japanese speakers participated in each subjective evaluation. Each system was evaluated over 270 times.

## 4.3. Results for generalization of latent space variables

To verify the degree of speaker independence of the latent space variables, we calculated the differences between the reconstruction error and the conversion error for each method, which are shown in the second row of Table 2 (denoted by $(\cdot)$). If the latent space features are less speaker-dependent, the difference between the reconstruction and conversion errors should be small. Conversely, if the difference is large, the latent space variables contain speaker information, which may have caused the conversion error to be larger.

The objective evaluation results show that the proposed

Table 2: *Objective evaluation results. The lower the MCD and CER, the better the performance. The higher the $F_0$Corr, the better the performance. The values following ± indicate confidence intervals. The first terms of the second row for each method represent the reconstruction errors on the evaluation dataset. The second terms (represented by (·)) indicate differences between the reconstruction and conversion errors. The confidence intervals of the reconstruction error is omitted for brevity.*

| System | MCD ↓ | $F_0$Corr ↑ | CER ↓ |
|---|---|---|---|
| CVAE-Small | $6.76 \pm 0.04$ | $\mathbf{0.73} \pm 0.01$ | $7.5 \pm 0.27$ |
| | $5.20\ (1.56)$ | $0.84\ (\mathbf{0.10})$ | $5.3\ (2.2)$ |
| + beta | $6.62 \pm 0.03$ | $\mathbf{0.72} \pm 0.01$ | $13.5 \pm 0.35$ |
| | $5.81\ (0.81)$ | $0.82\ (\mathbf{0.10})$ | $10.4\ (3.1)$ |
| + cc | $6.78 \pm 0.04$ | $\mathbf{0.72} \pm 0.01$ | $10.6 \pm 0.32$ |
| | $5.40\ (1.38)$ | $0.84\ (0.12)$ | $7.1\ (3.5)$ |
| + ac | $6.75 \pm 0.04$ | $\mathbf{0.72} \pm 0.01$ | $\mathbf{7.2} \pm 0.26$ |
| | $5.18\ (1.57)$ | $0.85\ (0.13)$ | $4.9\ (2.3)$ |
| + pr (ours) | $\mathbf{6.57} \pm 0.03$ | $\mathbf{0.73} \pm 0.01$ | $9.2 \pm 0.29$ |
| | $5.79\ (\mathbf{0.78})$ | $0.83\ (\mathbf{0.10})$ | $7.9\ (\mathbf{1.3})$ |
| CVAE-Large | $7.45 \pm 0.04$ | $0.50 \pm 0.01$ | $17.6 \pm 0.52$ |
| | $4.58\ (2.87)$ | $0.86\ (0.36)$ | $3.5\ (14.1)$ |
| + beta | $6.94 \pm 0.04$ | $0.62 \pm 0.01$ | $12.6 \pm 0.34$ |
| | $5.50\ (1.44)$ | $0.85\ (0.23)$ | $6.8\ (5.8)$ |
| + cc | $7.24 \pm 0.04$ | $0.59 \pm 0.01$ | $14.3 \pm 0.42$ |
| | $5.02\ (2.22)$ | $0.85\ (0.26)$ | $4.5\ (9.8)$ |
| + ac | $7.21 \pm 0.04$ | $0.60 \pm 0.01$ | $13.5 \pm 0.43$ |
| | $4.79\ (2.42)$ | $0.86\ (0.26)$ | $3.8\ (9.7)$ |
| + pr (ours) | $\mathbf{6.52} \pm 0.03$ | $\mathbf{0.73} \pm 0.01$ | $\mathbf{6.2} \pm 0.23$ |
| | $5.52\ (\mathbf{1.00})$ | $0.84\ (\mathbf{0.11})$ | $4.8\ (\mathbf{1.4})$ |

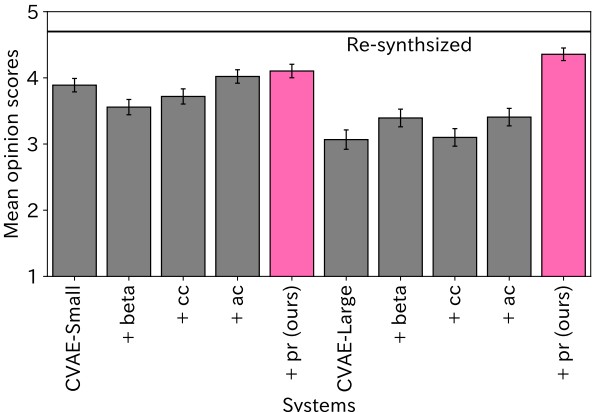

Figure 3: *Subjective evaluation results on sound quality. The higher the value, the better the sound quality. The error bars denote 95% confidence intervals. Re-synthesized indicates the speech synthesized from the ground-truth Mel-spectrogram.*

method (**+pr**) has the smallest difference, indicating that the latent space variables are more generalized. Moreover, when the model size is increased, the proposed method improves the reconstruction and conversion errors compared to the case with a smaller model size. In contrast, the conventional method has a smaller reconstruction error but a larger conversion error. These findings support the claim that the proposed method can extract

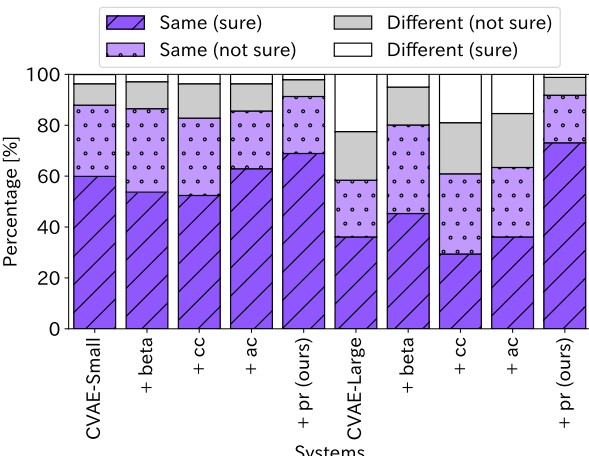

Figure 4: *Subjective evaluation results on speaker similarity. The higher the rate of Same, the better the performance.*

less speaker-dependent features as the latent space variables.

### 4.4. Results of subjective listening tests

Next, the results of the perceptual evaluation were shown in Fig. 3 and 4. From these results, the proposed method (**+pr**) with the large-sized model is the best system for sound quality and speaker similarity. Similar to the objective experimental results, the difference between the methods is smaller when the model size is small. However, as the model size increases, the conventional method deteriorates while the proposed method improves. Moreover, while the beta variant (**+beta**) performs less well than other methods when the model is compact, it outperforms other conventional methods when the model is large. These results suggest that constraints on latent space variables have a certain effect.

## 5. Conclusions

This paper described a non-parallel many-to-many voice conversion method based on a perturbation-resistant variational autoencoder. We introduced an encoder trained to match the encoded features of the input speech with those of a pseudo-speech generated through a content-preserving transformation of the input speech's fundamental frequency and spectral envelope. Experimental results showed that the proposed encoder enabled us to extract less speaker-dependent features, leading to the best performance in subjective evaluations. We plan to extend the proposed speech representation technique to other downstream tasks, such as automatic speech recognition and source separation.

## 6. ACKNOWLEDGMENTS

Empty for double-blind review. This work was supported by JST CREST Grant Number JP-MJCR19A3, Japan.

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
