# OpenReview forum: "PRVAE-VC: Non-Parallel Many-to-Many Voice Conversion with Perturbation-Resistant Variational Autoencoder"
_Interspeech.org/2023/Workshop/SSW — SSW12_

### Official Review · Reviewer_rdLq · 2023-05-30
**Better non-parallel VC by making VAE robust to content-preserving permutation**

**Rating:** 7
**Confidence:** 5

**Review:**

- Summary: This paper proposes a VAE-based non-parallel VC model which can robustly extract speaker-independent, content representation to achieve better voice conversion performance.  They added a loss between latents from original speech and perturbed speech modified via content-preserving, signal-processing-based modulation.  This loss encourages the encoder to extract less speaker-dependent, content representation from input speech and improves the VC performance.  The VC performance was evaluated in different metrics such as ASR, MCD, F0 correlation, and subjective metrics.

- Quality: Good

- Clarity: Good

- Originality: Good

- Pros: Good performance and convincing results.

- Cons: Small evaluation (ATR 503 dataset).  Was evaluation conducted with unseen speakers as input?  Or does it work only with seen speakers?  The paper says "10-speaker input and 10-speaker output", but not mention which speakers / speaker combinations were used in the evaluation.

---

### Official Review · Reviewer_aD4b · 2023-06-02
**Please conduct directly objective tests for speaker independency of latent variables.**

**Rating:** 7
**Confidence:** 4

**Review:**

## Summary of the paper
This paper challenges the reduction of speaker dependent features from latent space variables in VAE-based non-parallel voice conversion.
To reduce them, authors have proposed Perturbation-resistant VAE framework, which introduces modified analysis-synthesized speech as perturbation training data and considers losses (called as perturbation resistant loss in this papper) between latent variables of original data and those of perturbation data.
In experimental evaluations, authors have compared the proposed method to several conventional VAE-based VCs (conditional-, beta-, cyclic-consistent-, and auxiliary classifier-) in the some aspects.
Experimental results show that the proposed method, especially the proposed large model, has better performace than any other conventional methods.

## Key Strength of the paper
- This paper proposes a novel loss function which is called "perturbation resistant loss" in order to remove speaker bias from latent space variables.
- Experimental results show that the proposed VAE-based VC outperforms other ones.

## Main Weakness of the paper
- Authors have not evaluated speaker-independencies of the latent space variables directly in experimental evaluations.



## Novelty/Originality
- This paper directly calculates differences between original and perturbation latent variables and optimizes model parameters.
  This idea is very simple and enough effective for improving model performance.


## Technical Correctness
- Authors provides details of model architectures, training conditions and corpus information used in experiments in this paper. So, we can reproduct experiments.


## Suggestions for improvement
- Pointed out above, authors have not evaluated speaker-independency of the latent space variables.
Of course, authors show the perturbation resistant losses in each iteration in figure 2, but I guess that these are calculated from same speakers' original and purterbation data.
To verify that the proposed method can remove speaker biases from latent variables more clearly, you should calculate above loss or other evaluation criteria between different speakers, and show their results.
Fortunately, authors uses ATR database set, which consists of parallel dataset and so you can calculate the perturbation resistant loss between different speakers by using time warping functions calculated in mel-cepstral distance.

- In Section 4.3, Authors indicate that the purpose of objective evaluation is to verify the speaker independency of the latent variables. However, these results are affected by not only encoders' but also decoders' performance. Therefore, I consider these evaluations are not enough to achieve authors' purpose, and the discussions in this section should be  only referred to objective performance of the VC models.

- In Columns of F0 Corr., of table 2, all upper values are written by bold font. I think "+beta", "+cc" and "ac" should be written by normal font.

## Quality of References
References are adequate.

## Clarity of Presentation
I can understand descriptions easily.

---

### Decision · Program_Chairs · 2023-06-14

**Decision:**

Accept

**Comment:**

SSW2003 received 45 papers. The acceptance rate is 82%. We are pleased to inform you that your paper has been accepted by the SSW2023 Program Committee. Please read the reviews carefully and submit your camera-ready paper by June 28th. Most reviewers performed a detailed review. Please answer to their questions and consider their comments. Note that camera-ready papers are credited with one extra page to allow authors to consider reviewers’ suggestions. So max 7 pages in total including figures & refs.
The deadline for submitting the revised version (with full non-anonymized authors and refs!) is 28th June.